# Nutritional Support in Pancreatic Diseases

**DOI:** 10.3390/nu14214570

**Published:** 2022-10-31

**Authors:** Pablo Cañamares-Orbís, Guillermo García-Rayado, Enrique Alfaro-Almajano

**Affiliations:** 1Gastroenterology, Hepatology and Nutrition Unit, San Jorge University Hospital, Martínez de Velasco Avenue 36, 22004 Huesca, Spain; 2Digestive Disease Department, Lozano Blesa University Clinic Hospital, San Juan Bosco Avenue 15, 50009 Zaragoza, Spain; 3Aragón Health Research Institute (IIS Aragón), San Juan Bosco Avenue 13, 50009 Zaragoza, Spain

**Keywords:** chronic pancreatitis, acute pancreatitis, pancreatic exocrine insufficiency, pancreatic cancer, malnutrition, oral nutritional supplements, enteral nutrition, parenteral nutrition

## Abstract

This review summarizes the main pancreatic diseases from a nutritional approach. Nutrition is a cornerstone of pancreatic disease and is sometimes undervalued. An early identification of malnutrition is the first step in maintaining an adequate nutritional status in acute pancreatitis, chronic pancreatitis and pancreatic cancer. Following a proper diet is a pillar in the treatment of pancreatic diseases and, often, nutritional counseling becomes essential. In addition, some patients will require oral nutritional supplements and fat-soluble vitamins to combat certain deficiencies. Other patients will require enteral nutrition by nasoenteric tube or total parenteral nutrition in order to maintain the requirements, depending on the pathology and its consequences. Pancreatic exocrine insufficiency, defined as a significant decrease in pancreatic enzymes or bicarbonate until the digestive function is impaired, is common in pancreatic diseases and is the main cause of malnutrition. Pancreatic enzymes therapy allows for the management of these patients. Nutrition can improve the nutritional status and quality of life of these patients and may even improve life expectancy in patients with pancreatic cancer. For this reason, nutrition must maintain the importance it deserves.

## 1. Introduction

Pancreatic gland is a doubly important organ, because it has two momentous functions. Pancreas is involved in digestion and in hormone secretion, including insulin and glycemic regulation. To understand and study, the pancreas has been academically divided in exocrine and endocrine pancreas. Actually, both pancreases form a single organ and are closely related each other.

From a nutritional point of view, exocrine function covers more importance. Changes in this function could observe in acute or chronic pancreatic injury, and in patients with pancreatic cancer. This review focus in the nutritional alterations caused by pancreatic diseases and try to unify current solutions.

## 2. Malnutrition Definition and Assessment

Malnutrition is the consequence of a lack of intake or uptake of nutrients, which causes an alteration in body composition, resulting in physical and mental deterioration [1].

Carrying out a nutritional screening is a fundamental intervention in the diagnosis of any pancreatic disease and at regular intervals during therapy. According to The Global Leadership Initiative on Malnutrition, a two-step strategy is recommended to evaluate malnutrition. The first step consists of identifying patients at risk of malnutrition in order to subsequently confirm the diagnosis and assess the severity in a second step [2].

Different nutritional screening tools have been vetted for inpatient and outpatient use to detect malnutrition. The Malnutrition Screening Tool (MST) is a quick, easy to use and reliable screening tool. It consists of two questions, one concerning unintentional weight loss and one concerning decreased appetite. According to the total score obtained, patients can be classified as being at low (0 points), moderate (2 points) or high risk (3–5 points) of malnutrition [3]. Another validated, quick and easy screening tool is the Malnutrition Universal Screening Tool (MUST). It is a five-step score that evaluates body mass index (BMI), weight loss and the effects of acute illness. Finally, patients could be differentiating in low risk (MUST 0 point), medium risk (MUST 1 point) or high risk (MUST ≥ 2 points) of malnutrition [4].

According to some consensus documents [5], after the stratification of the patients using the screening tools, the subsequent approach can be decided by the patient. Low risk patients could be controlled and followed by the gastroenterologist, oncologist or surgeon. Medium risk patients could be controlled by the doctor in charge of treatment, but a nutritional counselling is recommended. High risk patients must be referred to a hospital’s Nutrition Unit. The presence of a nutritionist in the multidisciplinary pancreatic unit is recommended for early management of malnutrition.

Another tool for early detection of malnutrition is through biochemical parameters that must be frequently monitored in patients with pancreatic disease, mostly in chronic affectation. It is common to detect decreased levels of vitamin A, vitamin D, vitamin E (englobe as fat-soluble vitamins), retinol transport protein, prealbumin, osteocalcin and essential minerals such as magnesium, zinc or selenium in these patients [6,7].

## 3. Chronic Pancreatitis

### 3.1. Definition, Etiology and Diagnosis

Chronic pancreatitis (ChP) is a fibroinflammatory disease characterized by the replacement of the pancreatic gland with fibrotic tissue because of repetitive inflammation of the pancreas [8]. It is a dynamic and progressive process, and the clinical symptoms and signs crop out as fibrosis develops. The most frequent symptom is abdominal pain in the upper abdomen, and it is often worsened by food intake [9], although it does not always correlate with the extension of pathological changes of ChP [10].

Alcohol abuse is the most frequent risk factor for the development of ChP. Some authors suggest that the amount and duration of alcohol consumption is 80 g per day for a period of at least six years [11]. Often related to the above-mentioned, smoking is a dose-dependent risk factor [10]. Genetic mutations, cystic fibrosis, autoimmune, hypercalcemia or obstructive repeated pancreatitis are other alternative causes of ChP. However, idiopathic ChP accounts for up to 28% of all the etiologies [10]. 

Accurate diagnosis of calcifying or advanced ChP is relatively simple. Compatible symptoms and calcifications on computed tomography scan or magnetic resonance imaging may be enough. However, diagnosis in the early stages is a difficult challenge. Inconclusive changes might be observed radiologically. Although endoscopic ultrasound might be helpful, it does not have enough accuracy for early stages [12]. The presence of factor risks, the adequate clinical features and high suspicion remain more important. Performing a pancreatic biopsy for diagnosis is not a viable option in clinical practice [13]. Ideally, the diagnosis of ChP should be made prior to the development of exocrine pancreatic insufficiency [13]. Some endoscopic techniques, such as ultrasound elastography, have shown promising outcomes to support diagnosis of ChP [14,15], but they are limited to tertiary centers with experience, and remain under investigation. From the radiological side, T1 relaxation time of pancreatic parenchyma might also be able to detect early changes in ChP [16].

### 3.2. Pancreatic Exocrine Insufficiency

Pancreatic exocrine insufficiency (PEI) and ChP are not the same concept. PEI occurs mainly in the context of ChP as the pancreatic parenchyma is affected, theoretically, in more than 90% [17]. PEI is defined as the pathological decrease in enzyme secretion (mainly lipase but also amylase, protease, etc.) or bicarbonate [18], causing an insufficiency in realizing the digestive function. For this reason, there are more circumstances, in addition to ChP, in which PEI can also develop: (1) after pancreatic resection surgeries; (2) temporarily after necrotizing acute pancreatitis until parenchyma is recovered; (3) in pancreatic cancer affecting the head or neck; (4) in partial or total gastrectomy or in Zollinger-Edison syndrome, because of an inactivation of lipase due to an acidic environment in the duodenal lumen [17].

According to a cross-sectional and multicentered Spanish study, the rate of PEI in patients with ChP was 64.1%, measured through fecal elastase [19]. Other studies have published rates of 84.6%, based on clinic [20].

The main clinical sign of PEI is steatorrhea, although this may be a sign of an advanced stage. However, the absence of steatorrhea does not indicate a correct absorption of fat and other nutrients. Quantification of fecal elastase in feces is the most widely used method to assess the enzymatic secretion, because elastase is a very stable enzyme formed almost exclusively in the pancreas. However, there are other direct and indirect measurement methods that are not the subject of this review [17].

Fecal elastase shows, in general, adequate sensitivity, specificity and predictive values; levels above 200 µg/g are considered normal, while levels below 100 µg/g indicate PEI. Values between 100 µg/g and 200 µg/g are indicative of possible PEI but must be evaluated in the general context of the patient [21]. Fecal elastase has a good negative predictive value, meaning the diagnosis of PEI can be ruled out in patients with a low probability, such as those being studied for diarrhea, when elastase is above 200 mcg/g [22]. However, fecal elastase has some limitations: the determination in liquid faeces can give false positives; it is unsuitable for patients who have undergone a pancreatoduodenectomy; and, if pre-test probability of PEI is not high, fecal elastase does not have enough of a positive predictive value, so a value below 200 mcg/g should not be attributed directly to PEI and pancreatic parenchymal must be evaluated [23]. Other pathologies such as celiac disease, inflammatory bowel disease, diabetes mellitus or bile acid diarrhea have also been linked to low levels of fecal elastase [23].

In the context of ChP, a Spanish study compared fecal elastase with the gold standard test, the coefficient of fat absorption, to evaluate the diagnostic capacity of PEI [24]. The results showed a good sensitivity above 80% for any value of elastase, an ascending specificity (up to 80%) if elastase levels are lower, a positive predictive value below 70% and, as previously mentioned, a good negative predictive value above 90%. The best cutoff level for a better discrimination was 84 µg/g, achieving a sensitivity of 87.5%, a specificity of 81.6%, a positive predictive value of 66.7% and a negative predictive value of 93.9% [24].

### 3.3. Chronic Pancreatitis and Malnutrition

There are two mechanisms that can primarily produce malnutrition in ChP. Firstly, the lower absorption of nutrients caused by the decrease in enzymes secretion, once PEI has been established. Secondly, patients progressively reducing their caloric intake, due to the fear of abdominal pain after eating [25]. Moreover, the effect of alcohol abuse and smoking may impoverish the nutritional status.

The use of previously mentioned scores to perform a malnutritional screening (MUST or MST) is important [17,26,27]. The rate of patients with ChP who present malnutrition is around 30%, according to a study using the MUST score [20]. On the other hand, several prospective studies comparing patients with ChP, and healthy controls have described a rate of close to 50% for patients with ChP who are either overweight or obese [6,27,28]. It is necessary to emphasize that overweight patients can also be malnourished. Physicians should not rule out the diagnosis of ChP based on patient’s phenotype alone, if complementary tests are conclusive. However, patients with low BMI have more severe ChP and lower quality of life [29].

As mentioned above, a frequent consequence of malnutrition is the reduction in serum fat-soluble vitamins such as vitamin A, D, E and K, and other micronutrients in patients with ChP. Specific recommendations for their supplementation do not exist because of there are few studies evaluating this point [11].

Osteopenia and osteoporosis are other consequences of malnutrition. They are different degrees of the pathology of bone metabolism. Osteopenia and osteoporosis are measured by a T-score: bone density compared to the peak bone mass, used for young adults; or by a Z-score: bone density compared to the population mean according to gender and age. Osteopenia refers to a lesser decrease in bone density, and osteoporosis to a deeper decrease in bone mass. It is usually measured by dual-energy X-ray absorptiometry (DXA). Although ChP can worsen vitamin D absorption, not only vitamin D levels influence it. Other significant influences include smoking, low physical activity, and chronic inflammation at the systemic level, secondary to ChP [25]. Up to 65% of patients with ChP present osteoporosis or osteopenia [28]. Taking only osteoporosis into account, the prevalence is 23.4% according to a meta-analysis in which DXA was used as the diagnostic tool [28]. 

However, the rate of bone fracture is clinically more important than the prevalence of osteoporosis in patients with ChP. Several retrospective studies, with a large sample size, have shown an increase between 2.4 and 3 times greater than in the control group [30,31]. Osteocalcin, as a marker of osteoblastic activity, has also been shown to be decreased in serum in patients with ChP [6]. As a result, DXA may be used more frequently, and earlier, to detect patients with an increased risk of fractures.

### 3.4. Optimal Diet in Chronic Pancreatitis

A low-fat diet has conventionally been recommended to patients with ChP, because it was thought that the pancreas would rest and, consequently, that the steatorrhea would improve. In fact, a low-fat diet decreases pancreatic secretion [32], and therefore the energy from fat intake decreases, in addition to malabsorption of carbohydrate and protein, secondary to the lower pancreatic enzyme secretion. 

Currently, a balanced and healthy diet without fat restriction is recommended in clinical guidelines [11,25,26]. Normal nutritional requirements depend on weight and physical activity. Patients with ChP may need higher requirements to balance the potential malabsorption. For example, about 2100 Kcal are recommended for a person with 70 kg of weight (calculating about 25–35 Kcal/kg), so an intake of 2800–3000 Kcal would be recommended if this person had ChP. Up to 30–40% of energy given as fat is well tolerated, especially if it is vegetable fat [33]. A high protein-diet (around 1–1.5 g per kg body weight per day) must be suggested for patients with poor nutritional status and a frequent intake of between 5–6 meals a day should be prescribed [25].

In addition, a diet with low or moderate fiber intake is appropriate, because fiber can reduce the effectiveness of supplemental enzyme treatment and cause malabsorption [25,26].

These recommendations should be accompanied by the cessation of alcohol and tobacco consumption. Weight reduction should be recommended in patients with CP and obesity, if severe malnutrition and sarcopenia have been excluded [27].

A diet similar to the Mediterranean diet is a well-balanced alternative for ChP patients due to its high content of fruits, legumes, nuts, cereals, fish and unsaturated fats (such as olive oil). Another important element in the Mediterranean diet is polyphenols [34]. They may have a role in the improvement of some inflammatory diseases through the regulation and management of reactive oxygen species (ROS) and immunomodulation, although the specific evidence in regard to pancreatic diseases is limited [35]. Due to the risk of micronutrient insufficiency, indiscriminate vitamin supplementation for individuals with ChP is not advisable, and individual evaluation should be conducted, including dietary intake [6]. It is recommended to frequently measure biochemical levels, and to give replacements in those patients with low serum levels, and no-responder to diet as an individualized decision in each patient [6]. Regarding vitamin D, there is a study that showed oral vitamin D (1520 IU per day) was significantly more effective at increasing serum 25-hydroxyvitamina D than UV radiation in patients with ChP [36]. Vitamin D supplementation should be considered in patients with osteopathy (osteopenia and osteoporosis), especially if a history of bone fractures is presented [6].

Both a counseled-home-made diet and nutritional supplements improve anthropometric parameters, fecal fat excretion and pain scores, according to a randomized clinical trial [37]. If an adequate caloric intake or weight gain is not achieved with the diet, expert dietary advice would be the best option. When a clinical nutritionist is not available, nutritional supplements are a good alternative. Formulas with medium chain triglycerides do not seem to be more effective than the alternatives, however, there are no randomized trials comparing the different supplement formulas [25]. Approximately 10–15% of patients with ChP will need nutritional supplements at certain times [38].

### 3.5. What Happens When Oral Nutrition Is Insufficient?

There are data indicating that approximately 5% of patients with ChP require enteral tube feeding nutrition [39]. The indication recommended in the clinical practice guidelines is for those patients who require rapid weight gain or who present significant abdominal pain with the intake of a normal diet [25]. This last group includes patients with a poor tolerance, due to duodenal stenosis or persistent nausea, for delayed gastric emptying.

Studies comparing nasogastric (NG) and nasojejunal (NJ) tubes are multiples in acute pancreatitis (AP), showing that a NG tube is safe and well-tolerated in most patients, as will be explained later [40]. Theoretically, this might also be assumed for ChP. However, no studies have been performed comparing both enteral tube access, so the evidence in ChP is scarce. Currently, experts advise the use of a NJ tube for these patients [41], or, in case the nutrition is expected to last more than 30 days, experts advise the placement of a jejunostomy, through a gastrostomy or a direct percutaneous jejunostomy, that can be placed safely and effectively by balloon enteroscopy [42]. This recommendation has stronger support in cases where a patient suffers from intense abdominal pain or persistent nausea. 

Elemental nutritional formulas are capable of reducing pain episodes in patients with ChP, according to some studies [43,44]. Theoretically, semi-elemental enteral nutrition could also be effective, but there are no studies comparing the different nutrition formulas in ChP frame [45].

### 3.6. Pancreatic Enzyme Replacement

Detecting the presence of PEI is one of the first actions when ChP is diagnosed in a patient. Pancreatic Enzyme Replacement Therapy (PERT) should begin when compatible clinical signs, weight loss or biochemical markers of malnutrition occur [6,25]. Fecal elastase can be used to support the diagnosis, but it should not be the unique altered value.

All pancreatic enzymes come from a porcine origin. Amylase, lipase, protease and nucleic acids are contained in microspheres that are covered by an enteric capsule to resist the stomach acidic aim, and microspheres are released when the pH is greater than 5.5 in the duodenal lumen [25]. PERT should be taken during or after meals to achieve a better distribution with the chyme formed in the stomach [46]. The recommended dose of PERT is 40,000–50,000 Ph.U. (units refer to amount of lipase) with each meal and 25,000 Ph.U. with snacks [47]. A decrease in fecal fat excretion, fecal weight, and abdominal pain, without major side effects, has been shown with the use of PERT in a meta-analysis published in 2017 [18]. The dose should be adjusted according to the weight gain and normalization of biochemical markers [18,25]. There is no maximum dose because what is not used is eliminated through the digestive tract.

When patients with ChP undergoing PERT are compared with those who are not, the former present increased levels of vitamin A and prealbumin [6].

### 3.7. How Pancreatic Function Can Be Measured in Patients on Enzymatic Treatment?

Fecal elastase cannot be used to assess the response to PERT because the treatment does not modify the intrinsic elastase formed by the patient [17]. Alternative diagnostic tools for measuring the treatment response are the ^13^C-mixed-triglyceride breath test and the coefficient fat absorption, also called Van Kamer test, although these tests are no longer readily available in most hospitals.

If the initial response to treatment is insufficient, there are two options to consider (Figure 1). The first is to optimize the treatment by adding prompt pump inhibitors (PPI) to reduce gastric acidity and to ensure that enzyme release occurs correctly in the duodenal lumen. The second option is to increase the dose of enzyme units, since the physiological secretion is much greater, and the necessary dose is probably higher [48]. However, the first option has contradictory data. The favorable data are obtained from an experimental non-randomized clinical trial carried out prospectively in 21 patients, in whom fat absorption was assessed by means of the ^13^C-mixed-triglyceride breath test before and after taking PERT and PPI [48]. Patients’ responders to PERT did not improve when PPI was added to treatment, however, patients’ non-responders to PERT led to an absolute improvement in fat digestion of 14.3% (95% CI 3.4–47.6%). In contrast, a retrospective analytical study subsequently carried out with data from 1142 patients participating in different clinical trials did not observe any benefit in acid suppression using PPIs or histamine antagonists [49]. Despite these results, this recommendation remains in most clinical practice guidelines [11,25].

## 4. Acute Pancreatitis

### 4.1. Introduction

AP is the third gastrointestinal disease that requires a greater number of hospital admissions [50], with an incidence estimated between 14–45 cases per 100,000 persons per year. Annual healthcare expenditures in the United States for AP totaled $2.7 billion [51]. In addition, with the advancing age in Western countries and increasing rates of obesity, the incidence of gallstone AP is markedly increasing in some countries [52]. AP is a heterogeneous disease; around two-thirds of AP patients have a mild course of illness with a fast improvement, however, one-third of AP patients can develop local complications (necrosis and/or fluid collection that can be infected) and/or organ failure, associated with significant morbidity and even risk of mortality. Overall, 25% of patients will have one or more recurrences of the disease [53]. Despite this, there is no specific treatment for AP, and the cornerstones of early management are fluid resuscitation, analgesia and nutritional support [54]. 

In AP, especially in moderately-severe and severe AP, the inflammation and septic complications result in an increase in metabolism, so energy requirements can be notably higher [55]. Energy expenditure may be 139% of that predicted by the Harris-Benedict equation [56]. In this context, protein catabolism is also increased, alongside an important protein loss (the nitrogen loss could be between 20–40 g per day). On the other hand, in AP, alterations in the water-electrolyte and acid-base balance can be developed, and many patients have a decrease in vitamins and micronutrients [55]. Furthermore, stress hyperglycemia from transient insulin resistance and decreased insulin production of inflamed islet cell occurs in many patients with AP [56]. Thus, all patients with AP are at risk of malnutrition and nutritional support is required.

### 4.2. Gut Rousing Theory and No Pancreatic Rest

Historically, patients hospitalized for AP fasted (nil per os—NPO) in order to minimize pancreatic stimulation, which could exacerbate pancreatic inflammation, according to the “pancreatic rest” theory. Subsequent studies have shown that exocrine pancreatic function is decreased in patients with AP and, furthermore, this decrease is inversely proportional to the degree of severity of AP (the greater the severity, the greater the deterioration of the exocrine pancreatic function) [57]. Thus, oral or enteral nutrition (EN), regardless of the route, does not significantly increase pancreatic secretion in AP and it does not worsen pancreatic inflammation; in fact, it has a beneficial effect, by promoting intestinal integrity and functionality and by decreasing inflammatory response [58]. Oral or EN decreases gut permeability by holding the functional tight junctions between the intestinal epithelial cells closed, increasing blood flow to the gut and promoting the release of immunoglobulin A and bile salts [56]. 

Finally, many studies have shown that oral/EN (compared to NPO) in patients with AP improves the course of the disease by reducing local complications (infection of necrosis), organ failure and mortality [59,60]. For all of these reasons, the “pancreatic rest” theory has now been ruled out and has been replaced by the “gut rousing” theory.

### 4.3. Nutrition in Mild Acute Pancreatitis

Patients with mild AP are characterized by the absence of organ failure and local complications and present rapid clinical improvement with decrease/cessation of abdominal pain and absence of systemic inflammatory response syndrome (SIRS). In these patients, several studies have shown that an early reintroduction of oral feeding is safe, well tolerated (not result in a relapse of symptoms) and shortens the length of stay (LOS) in hospital. For example, in a randomized control trial (RCT) published in 2007, 60 patients were randomized to immediate oral refeeding or NPO and subsequent gradual reintroduction of the diet. Patients with immediate oral refeeding did not experience a worsening of intestinal symptoms and had a shorter LOS (4 days vs. 6 days, *p* < 0.05) [61]. In addition, oral feeding can be reintroduced with a solid diet without the need to start with a diet of clear liquids, and then progress to a solid diet. In this sense, several RCTs have concluded that direct oral refeeding with a solid diet, instead of a diet of clear liquids, provides more calories, does not worsen symptoms and can shorten LOS [62,63].

### 4.4. Nutrition in Moderate-Severe Acute Pancreatitis

#### 4.4.1. Timing

The timing of beginning nutrition in patients with predicted severe AP has been subject to controversy in recent decades, with studies showing discordant results. The evidence is based on observational studies, RCTs, and meta-analyses. As for observational studies, some retrospective analysis has shown positive results when starting EN within the first 48–72 h. These studies concluded that early EN in patients with AP with predicted severity was associated with a decrease in pancreatic infection, respiratory and renal organ failure, and mortality [64,65]. Along the same lines, a RCT that included 60 patients with severe AP suggested a decrease in SIRS, organ failure and pancreatic infection in patients for whom EN was started in the first 48 h, compared to postponing it until the 8th day. However, this RCT did not show a decrease in mortality in these patients [66]. In contrast, a RCT published in 2016, with 214 patients with predicted severity, did not show a decrease in SIRS, persistent organ failure, or mortality when EN was started within the first 24 h [67]. The evidence from meta-analyses is also not homogeneous. Two meta-analyses concluded that EN in the first 24 h and 48 h, respectively, decreased complications in patients with AP with predicted severity [68,69]. However, a more recent meta-analysis showed no significant difference in outcomes when starting EN in the first 48 h in predicted severe AP [70]. Considering the beneficial results of some of the aforementioned studies, many recommendations and reviews on the subject have traditionally advised starting EN within the first 48 h in patients with predicted severe AP [55,71]. However, a RCT published in 2014 in the prestigious New England Journal of Medicine by the reference Dutch group (Dutch Pancreatitis Study Group) cast serious doubt on this recommendation [72]. This RCT included 208 patients with predicted severe AP (Acute Physiology and Chronic Health Evaluation II score ≥ 8, modified Glasgow score ≥ 3, or a serum C-reactive protein level ≥ 150 mg/L), who were randomized to EN by nasoenteric tube feeding within the first 24 h (early group), or to test tolerance to an oral diet 72 h after onset (on-demand group). In this last group, a NJ tube was provided to start EN if in the following 24 h, the patients did not tolerate oral feeding (therefore 96 h after presentation). There were no significant differences between the two groups in terms of mortality, infection, or development of organ failure. In addition, in the on-demand group, 69% of patients tolerated oral feeding and did not require a nasoenteric tube. As possible limitations, in the early group, 38% of patients detached from the feeding tube and some patients with predicted severe AP finally had a mild AP, which could have favored not finding differences between groups. However, these two possible limitations reflect the real clinical practice in these patients with AP. With the results of this RCT, some reviews and recommendations on the subject [58,73,74,75] do not suggest placing a tube for EN in the first 48 h in all patients with predicted severe AP, but recommend cautiously testing oral tolerance if feasible (good level of consciousness, absence of repeated vomiting, absence of intestinal ileus or intestinal obstruction), and suggest reserve EN by tube only for patients who do not tolerate oral feeding on the 4th day after presentation (Figure 2). 

In this regard, the 2018 American Gastroenterological Association guideline [76], although lacking a clear distinction between mild and severe AP, recommends oral feeding within the first 24 h, but recognizes that in some patients, due to pain, vomiting and gastrointestinal dysmotility, nutrition can be delayed beyond the first 24 h and that a significant proportion of patients tolerate oral diet even in severe/necrotizing AP, and EN can be reserved for patients who cannot tolerate oral feeding. Likewise, this guideline states that in AP with predicted severity and/or proven pancreatic necrosis, further studies are necessary to determine the exact timing for beginning oral or EN support.

#### 4.4.2. Enteral vs. Parenteral Nutrition

In patients with AP who do not tolerate oral feeding, treatment with nutritional support is indicated. This nutritional support may be implemented through EN by nasoenteric tube or total parenteral nutrition (TPN). Initially, decades ago, TPN was recommended according to the previously mentioned “pancreatic rest” theory, but subsequent studies have shown the superiority of EN over TPN in patients with AP. In 2006, a RCT was published which included 70 patients with predicted severe AP, who were randomized to EN by NJ tube or TPN within the first 72 h of the onset of symptoms. The EN group had fewer infectious complications, less organ failure, and lower mortality [77]. Another RCT published in 2010 randomized 107 patients with severe AP to EN or TPN within the first week of hospitalization. Patients treated with EN had less need for intervention and a lower percentage of sepsis of pancreatic origin, organ failure and mortality [78]. This evidence for RCTs has been confirmed in numerous meta-analyses designed in recent years. In 2010, a systematic review published in Cochrane included 8 RCTs and concluded that EN compared to TPN in AP is associated with less need for surgery, fewer septic complications, and less development of organ failure and mortality [59]. Other systematic reviews have shown similar results [60,79]. 

As mentioned above, beneficial results with EN are probably due to better maintenance of the intestinal barrier and functionality, which leads to less bacterial translocation, and therefore, fewer infectious complications and less SRIS. Additionally, TPN has been associated with increased catheter-related infections and increased metabolic disturbances. 

Taking into account the evidence described, the main AP guidelines [76,80] make a strong recommendation in favor of EN over TPN in patients with AP who require nutritional support, reserving the use of TPN only for patients who do not tolerate EN (Figure 2). According to the 2018 American Gastroenterological Association guidelines [76], due to the existing strong scientific evidence on the subject, no further studies are necessary comparing EN with TPN in AP.

#### 4.4.3. Route of Enteral Nutrition (Nasogastric vs. Nasojejunal)

EN by NJ tube has conventionally been recommended, instead of by NG tube, due to a theoretical benefit in the clinical course of AP, as an NJ tube could stimulate less pancreatic secretion and could pose less risk of respiratory aspiration. However, subsequent evidence has not confirmed such benefits. Some RCTs have been published on the subject [81,82], although they have methodological weaknesses, such as small sample size, lack of blinding, and great heterogeneity. These studies have not found significant differences between the two routes of EN in terms of mortality, infectious complications or intestinal symptoms. In addition, several meta-analyses have been carried out [40,83] that have not shown differences between administration of EN by the NG or NJ route. For example, a Cochrane review was published in 2020 that included five randomized RCTs and concluded that the evidence was insufficient for affirming superiority, inferiority, or equivalence between NG or NJ routes in patients with severe AP [40]. More and better designed studies are needed to clarify the best route of administration of EN in patients with AP [76]. 

According to the aforementioned studies, nutrition by NG tube seems to be feasible, safe, and well tolerated, and no differences have been found in the main outcomes between EN by NG or NJ tubes, so most recommendations suggest first trying the NG tube nutrition, as the placement of a NG tube is easier and cheaper compared to a NJ tube, which requires endoscopic placement. However, in the case of gastroparesis/delayed gastric emptying or gastric or duodenal obstruction from collections of AP, the preferred route would be NJ [84] (Figure 2). 

Finally, in patients who require EN for a long period (more than 30 days), according to general nutrition recommendations, a percutaneous gastrostomy or jejunostomy should be considered [85].

#### 4.4.4. Composition of Enteral Nutrition

Historically, elemental or semi-elemental EN formulations were preferred because they could have a benefit in patients with AP by less stimulating pancreatic secretion. Subsequently, several studies and meta-analyses have not found beneficial results when using elemental or semi-elemental formulas. A meta-analysis published in 2009, that included a total of 1070 patients with AP (825 were severe AP), found no significant differences in tolerance, infectious complications, or mortality [86]. Taking into account the absence of differences and the fact that polymeric formulations are cheaper, polymeric formulations are recommended for EN in patients with AP [58,71]. 

On the other hand, specialized formulations containing immunomodulatory supplements (mainly glutamine, arginine and omega-3 fatty acids) have been studied, which are known as immunonutrition. Some RCTs in the setting of critically ill patients have suggested positive effects (decrease in infectious complications and even mortality) by adding immunonutrition supplements to EN [87]. These effects have been less established in patients with AP. A Cochrane systematic review concluded that there were no significant differences in the development of SIRS and in mortality when using immunonutrition in EN in patients with AP [88]. Furthermore, most of RCTs included in this review had low quality and high risk of bias. Another meta-analysis also showed no beneficial effects of immunonutrition in patients with AP in terms of infectious complications or LOS [89]. Therefore, based on the existing evidence, the routine use of immunonutrition in patients with AP cannot be recommended. More studies with higher quality are necessary to clarify the possible influence of immunonutrition on the clinical course of AP. A new Cochrane systematic review whose protocol was published in 2019 is currently being developed [90].

The anti-inflammatory role of omega-3 is well-known; it functions by reducing proinflammatory eicosanoid and cytokine production [91]. However, studies about the supplementation in patients with AP are limited. There is a meta-analysis that includes 8 RCT that suggested a decrease in mortality, hospital stay and infectious complications in patients in whom omega-3 was used [92]. It should be noted that the predominant route of administration was the intravenous one that is less common (6 of these studies), and the dose varied in each study. The authors concluded that more randomized clinical trials are needed to reinforce the evidence. High-dose omega-3 could also reduce the triglyceride level [93], but the effect of omega-3 in the acute period of a hyper-triglyceridemic AP has not been evaluated, but has only been evaluated as an adjuvant treatment to reduce triglycerides levels after the acute condition.

## 5. Pancreatic Cancer

### 5.1. Introduction

Pancreatic cancer (PC) is one of the tumors with the worst prognosis. It represents only 3% of all cancers but is the sixth leading cause of cancer-related death in Europe. Its five-year overall survival rate is approximately 5% [94]. Temporary estimates of incidence and mortality predict a dramatic increase in the coming decades, of more than 70% [95]. 

Weight loss and malnutrition are highly prevalent problems in patients with PC, and contribute to low therapeutic tolerance, reduced quality of life and overall mortality. Up to 85% of patients present with weight loss at diagnosis and an additional 70% of patients developed malnutrition during chemotherapy [96,97]. 

Malnutrition, weight loss and cachexia are multifactorial and depend on tumor-related systemic factors (regulation of appetite, pro-inflammatory cytokines or increase in glycolysis and lipolysis), factors related to pancreatic function (pancreatic duct obstruction, disruption of lobular architecture or malabsorption) and gut- related factors (bacterial overgrowth or shift of gut microbiota) [98].

An appropriate approach to diagnosis and management of PC malnutrition is very important to improve different outcomes as quality of life and, possibly, overall survival.

### 5.2. Nutritional Screening

As previously noted, different nutritional screening tools (MST, MUST) must be used to detect malnutrition. Furthermore, the presence of a nutritionist in the multidisciplinary Oncology Unit is recommended for this early identification of malnutrition, as well as to help patients to maintain weight, better withstand the tumor treatment and maintain or improve their quality of life.

#### Assessment for Pancreatic Exocrine Insufficiency

A specific and very important aspect in relation to malnutrition in PC patients is the evaluation of PEI. As described above, PEI is a condition in which the production of pancreatic enzymes is insufficient, causing affected patients to develop malnutrition, owing to insufficient caloric intake and deficiencies of essential amino acid and micronutrients, such as lipophilic vitamins, iron or folic acid. PEI is a common condition in PC patients that affects two-thirds of patients with pancreatic-head tumors, and this proportion increases over the course of the disease, to more than nine out ten patients [99]. Tumor surgery, particularly pancreatoduodenectomy, is also highly associated with PEI [98].

The evaluation of PEI should occur at diagnosis, at the start of treatment and at regular intervals during the course of the disease.

However, a recent English consensus [100] suggests routine faecal elastase determination is not necessary in resectable and unresectable PC (mainly in tumors of the head of the pancreas, which are the most frequent), and recommends direct treatment because the presence of PEI at diagnosis is very high in patients with PC. Likewise, its appearance throughout the evolution of the disease is almost certain. Therefore, the benefit of performing diagnostic tests is poor given the high prevalence of PEI in patients with PC and the benefit of PERT, regardless of the result of fecal elastase that have been shown, in terms of nutrition and survival.

### 5.3. Nutritional Treatment

#### 5.3.1. Nutritional Counselling

Increased cell metabolism in cancer is one of the factors that can lead to cachexia. For this reason, nutritional interventions have long been emphasized as a way to improve cachexia related to cancer. Nutritional counselling is the first step; it provides patients adequate information about the dietary changes needed to achieve adequate calories and protein intake in their diet. This counselling must always be performed by a healthcare professional, and if possible, by a nutritionist with experience in the management of oncologic patients [5,101].

#### 5.3.2. Oral Nutritional Supplements

If patients show signs of deficiency, inadequate intake, or have prolonged malabsorption, clinicians should test for nutritional deficiencies such as vitamin A, D, E, K, B12 and other micronutrients (iron, zinc, selenium, copper). Serum values should be rechecked three months after replacement [102].

On the other hand, for patients with an inadequate intake of calories or protein, oral nutrition supplements are recommended as part of the nutrition care plan. They should be regularly controlled in order to evaluate effectiveness, and to control possible adverse effects, as well as to make the necessary dose adjustments as alternative therapies [103]. 

Whey proteins include different globular milk proteins, such as lactalbumins, leucine and lactoferrin, with the effect of improving the musculoskeletal system, cachexia syndrome, and increasing the insulin-like growth factors that could have a positive effect on tumoral cell apoptosis [104]. Other nutrients such as L-carnitine or branched-chain amino acids could play a role in cachexia, however, the evidence is still low [105,106].

#### 5.3.3. Enteral Nutrition and Parenteral Nutrition

Supplemental or total EN could be recommended for PC patients that cannot maintain an adequate oral intake despite oral nutritional supplements. This option is preferred when gut function is preserved. EN could be dispensed using a NG tube or by percutaneous endoscopic gastrostomy. Although patients with PC prefer gastrostomy over NG tube, the latter is related to lower complication rates [101]. The expected time to carry EN could help to choose between both alternatives: in the case of a shorter time, usually less than 30 days, the nasogastric tube is preferable; and for a longer time, the gastrostomy is more convenient [101]. 

TPN could be used when EN is insufficient or contraindicated. TPN may improve weight and maintain performance status in selected patients, but it is not routinely recommended. Careful evaluation should be used to identify those patients who would benefit it, and by other indicators, such as life expectancy; ethical aspects must be taken into consideration. For these reasons, prescribing TPN must be an individualized and multidisciplinary decision in the Oncology Unit, involving physicians and nutritionists [107]. 

If there is availability in the hospital, all-in-one multi-chamber TPN bags should be used. These could reduce the risk of infection, as well as reducing the preparation time in the Pharmacy Unit. To convert multi-chamber bags into complete ready-to-use all-in-one admixtures, pharmaceuticals with experience in this field are necessary [5].

#### 5.3.4. Pancreatic Enzyme Replacement Therapy

PERT has demonstrated benefits in patients with PEI due to pancreatic surgery or pancreatic cancer.

In a double-blinded trial of patients with unresectable PC, patients randomized to PERT had improved fat absorption, and weight loss was prevented compared to the placebo group [108]. In another retrospective and non-randomized study of unresectable PC patients receiving PERT and standard palliative care, or only standard palliative care, the median survival of patients receiving PERT was longer than those with standard palliative care (301 vs. 89 days) [109]. A prospective study of unresectable patients receiving chemotherapy showed changes in BMI and in the overall survival in patients who received PERT, compared with historical controls [110]. In addition, in a recent Spanish retrospective study, PERT appeared to improve survival in patients with unresectable PC [111]. Finally, another retrospective and observational study in the United Kingdom, with a large sample of patients, indicated an adjusted median survival time that was 262% longer among PERT patients compared with non-PERT patients [112]. In conclusion, PERT in PC seems to be associated with improved survival and nutritional status, although data should be prospectively validated in other studies, as the above-mentioned examples are retrospective, which may be a limitation.

The main problem with PERT is its underuse in patients with PC. In a retrospective study of 4554 patients with PC, only 21.7% had been prescribed PERT [112].

The initial PERT dose in ChP has previously been established as about 40,000–50,000 Ph.U., but it is thought that patients with PC might need more requirements from the beginning. A recent Spanish consensus suggests an initial dose in PC should be higher than in ChP, at around of 75,000 Ph.U. per meal, in PC patients [113]. It is also important to link PPI to treatment because patients with PC usually have reduced or abolished pancreatic secretion of bicarbonate.

#### 5.3.5. Nutrition Medical Treatment According to Pancreatic Cancer Status

The tumor status of PC is fundamental to its management, as well as in aspects related to nutrition. According to some consensus documents, different scenarios have been proposed based on National Comprehensive Cancer Network guidelines, in which PC is classified as resectable, borderline resectable, locally advanced non-resectable and metastatic [114].

For patients who are going to receive neoadjuvant treatment with chemotherapy, radiotherapy, or both, or who will receive adjuvant treatment, a nutritional evaluation is necessary before starting treatment and between sessions. It is also necessary to treat tumor-related or treatment-related symptoms such as anorexia, nausea and vomiting, pain or PEI, as these symptoms may affect the patient’s nutritional status [5].

If patients are candidates for surgery, a nutritional evaluation before surgery is also necessary, and if patients at risk of malnutrition are detected, nutritional supplements can be prescribed at least one week before surgery. For patients with severe malnutrition, the use of oral supplements, EN, or even TPN, can be considered to improve nutritional status before surgery. After surgery, an early onset of oral tolerance is recommended, and if postoperative complications do not allow oral feeding or cover patient nutritional requirements, the use of early TPN is an adequate strategy [5]. 

Finally, in cases of palliative patients, nutritional therapy must be individualized according to each patient, with the aim of achieving patient comfort and an adequate quality of life. For this purpose, multiple strategies can be used with the assessment of the Oncology and Nutrition Units. Currently, there is no clear consensus, so these can vary from the use of oral nutritional supplements, to EN, or in selected cases, the use of TPN [5].

## 6. Nutritional Support and Pancreatic Surgery

PC is often diagnosed in advanced stages. Only 20–30% of patients at the point of diagnosis are candidates for pancreatectomy [115]. The pancreas is a difficult-to-access organ to perform surgery on, due to its localization. Early surgical complications are related to pancreatic leakage, bleeding, delayed gastric emptying and local infection. Late surgical morbidities include exocrine, endocrine pancreatic insufficiency and afferent loop syndrome. Both of these can cause nutritional disorders: the former after the surgery, and the latter in the long-term [115].

However, malignancy is not the only indication for surgery. Some benign pathologies, such as advanced ChP when the management of abdominal pain becomes difficult or when some complications appear, such as duodenal stenosis or suspicious pancreatic mass, are also indications for surgery. The diagnosis of pancreatic cystic lesions has recent begun to increase. A small portion of these patients have high-risk signs, which will also require prophylactic surgery.

It is advisable to perform a meticulous evaluation of a patient’s nutritional status in the preoperative period. MUST scores have been associated with increased morbidity and mortality [116], and hypoalbuminemia is predictive of greater postoperative complications [116]. Currently, there is no supporting data for a widespread preoperative nutritional therapy. Guidelines recommend offering preoperative nutritional support only to patients with severe malnutrition [117]. This condition can also be recommended when one of the following criteria is present: (1) Weigh-loss of more than 15% in the previous 6 months; (2) a BMI less than 18.5 kg/m^2^; (3) serum albumin less than 30 g/L (without hepatic or renal disease) [117].

After the surgical procedure, EN is always preferrable due to its gut-barrier maintenance and possible reduction of infections. An observational study showed a decrease in hospital stay length when using an early oral feeding, without differences in morbidity [118]. Postoperative feeding tubes should be used with caution as they have been linked with postsurgical complications [116], however, the nasojejunal tube, placed by endoscopy, is preferable in necessary cases. Even in patients with pancreatic fistula, oral intake does not worsen the course [119]. On the other hand, TPN still has importance when oral tolerance is not available, for example in patients with delayed gastric emptying, paralytic ileus or anastomosis complications. When TPN is necessary, it should be started early, within three daysfollowing the procedure, as demonstrated by one clinical trial of patients undergoing abdominal surgery, which reduced nosocomial infections [120].

The start of PERT is routinely recommendable in patients after pancreatic resection [117]. The initial dose should be the usual dose, indicated previously in ChP patients for whom a pancreaticoduodenectomy or a distal pancreatectomy were performed. A higher initial dose may be necessary in patients undergoing total pancreatectomy. Fecal elastase may be not a good diagnostic marker because it will be predictably low after the surgery. For this reason, a high suspicion is important to detect PEI signs, such as steatorrhea, to adjust the PERT. Pancreaticoduodenectomy also has an increased risk of vitamin B12 and zinc deficiency [116]. However, the development of diabetes mellitus depends on the volume of resected pancreas, and it is not always present [116]. 

## 7. Conclusions

The nutritional status of patients with pancreatic diseases provides fundamental information. Many specialties are involved in the care of patients with pancreatic diseases, for example: primary care physicians, gastroenterologists, oncologists, pancreatic surgeons, endocrinologists. All specialists involved in the care of patients with pancreatic diseases should be aware of nutritional deficits in order to be detect it early, using high clinical suspicion and by performing nutritional scores and laboratory tests. If nutritional deficits are detected, these must be corrected through dietary advice, pancreatic enzyme replacement therapy and/or oral supplementation.

ChP is the main pancreatic disease related to malnutrition. Despite current difficulties, early ChP diagnosis will be improved by innovative techniques. Once the diagnosis is made, PEI must be studied based, usually, on indirect tests, such as fecal elastase, widely used due to its ease and availability. When PEI is detected, PERT should be initiated to try to maintain the nutritional status. However, patients with ChP may be malnourished for other reasons, so frequent screening, using nutritional scores such as MUST, and measurement of serum biochemical parameters, is recommended, both in patients with PEI and without PEI. Severe malnutrition should be managed by implementing an unrestricted, healthy, nutritional-counseled diet and an adequate calorie intake. Oral supplementation or, less commonly, nutrition through tube feeding, will be necessary for some patients.

AP is a frequent pancreatic disease which provokes a systemic inflammation. Mild AP may be managed with an early reintroduction of solid oral diet. However, nutrition in patients with moderate-severe AP are more difficult to manage. Oral feeding should be attempted within the first 72 h. When this is not possible, a NG tube feeding is advisable to provide the higher energy requirements. Some patients with delayed gastric emptying or duodenal obstruction do not tolerate NG tube feeding, so NJ tube is indicated. Finally, in some severe cases, TPN could be used if the patient is unable to receive nutrition by NJ tube.

PC is a disease that has a dramatic impact on life expectancy. Improvements for early diagnosis are also being investigated. Malnutrition depends on tumor-related systemic factors, but also on specific factors related to pancreatic function as PEI. Nutritional status assessment at the point of diagnosis and during the treatment of the disease is essential for the management of this type of cancer. The use of nutritional scores and serum biochemical parameters are recommended, although the use of fecal elastase can be avoided in certain circumstances. The use of PERT must begin earlier and at a higher dose as an increase in survival time has been suggested with the use of PERT. Ensuring adequate nutritional intake also improves the quality of life and the tolerance of chemotherapy in these patients.

## Figures and Tables

**Figure 1 nutrients-14-04570-f001:**
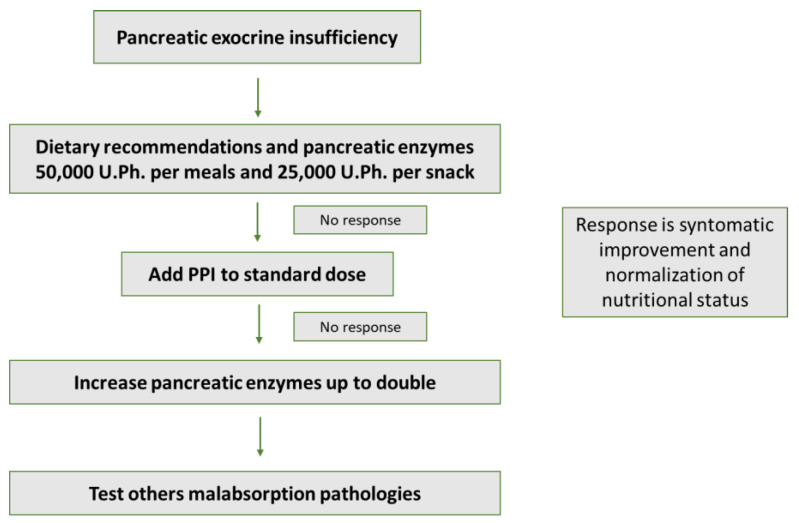
Management of PERT in pancreatic exocrine insufficiency.

**Figure 2 nutrients-14-04570-f002:**
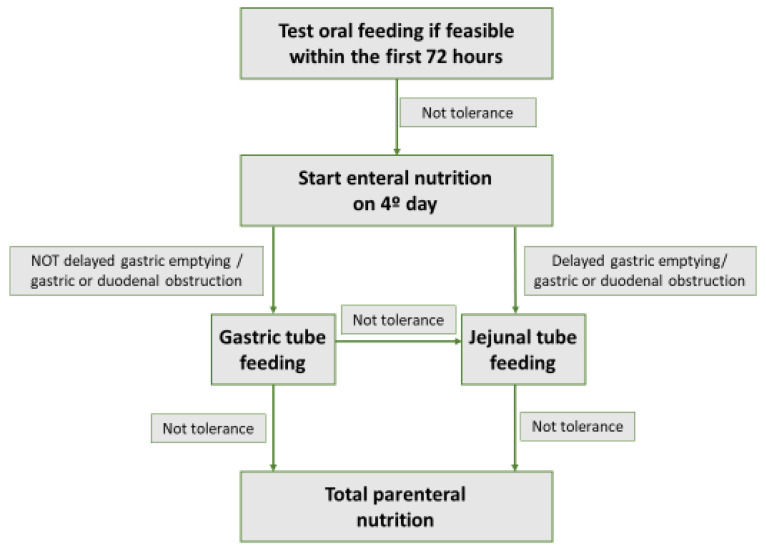
Nutrition algorithm in patients with predicted severe acute pancreatitis.

## Data Availability

Not applicable.

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
