# Peer review of "Nutritional Support in Pancreatic Diseases"

_nutrients, 2022, doi:10.3390/nu14214570_

Round 1
Reviewer 1 Report
The review by Pablo Canamare-Orbis and colleagues entitled “Nutritional Support in Pancreatic Diseases” is exhaustive and well written. The topic is of great interest in clinical practice and the review offers good practical counseling to manage pancreatic exocrine insufficiency and nutrition in pancreatic disorders.
Major Comment
If possible, it would be of interest to mention in the review also the nutritional support in post-duodenocephalopancreasectomy patients in order to complete the clinical scenario of pancreatic disorders.
Minor Comments
11. The images of the clinical cases are superfluous and can be omitted
22. The reading of the manuscript is a little laborious due to the editing errors throughout the manuscript: lack of parenthesis before and after n°of references, erroneous words division.
Reviewer 2 Report
In the cases presented, rather than radiographic images or CT scans, it would be interesting to report nutritional management.
As for cancer, vitamin A (as it could promote the cell cycle) as such should be avoided, perhaps consider beta-carotene, for the same reason B12, unless a compromised hematocrit, should be avoided.
Lactoferrin has a little scientific rationale, as does carnitine in particular; if the goal is to moderate cachexia, perhaps consider high BV proteins (such as whey) or leucine alone, but also, in this case, the action on protein synthesis could be adverse.
Instead, omega3 supplementation (in all three cases) should be considered as the anti-inflammatory action is well known.
As well as that of polyphenols, in particular, those typical of the Mediterranean diet (if not taken with food), as the modulating action on NFkB can be of sure support (see, for example; 10.3390 / antiox10020328)
Probably a better English check is needed.
